# Optimal community detection in dense bipartite graphs

**Julien Chhor**[*]
Department of Mathematics and Statistics
Toulouse School of Economics
Toulouse, France
julien.chhor@tse-fr.eu

**Parker Knight**
Department of Biostatistics
Harvard University
Boston, MA, USA
pknight@g.harvard.edu

## Abstract

We consider the problem of detecting a community of densely connected vertices in a high-dimensional bipartite graph of size $n_1 \times n_2$. Under the null hypothesis, the observed graph is drawn from a bipartite Erdős-Renyi distribution with connection probability $p_0$. Under the alternative hypothesis, there exists an unknown bipartite subgraph of size $k_1 \times k_2$ in which edges appear with probability $p_1 = p_0 + \delta$ for some $\delta > 0$, while all other edges outside the subgraph appear with probability $p_0$. Specifically, we provide non-asymptotic upper and lower bounds on the smallest signal strength $\delta^*$ that is both necessary and sufficient to ensure the existence of a test with small enough Type I and Type II errors. We also derive novel minimax-optimal tests achieving these fundamental limits when the underlying graph is sufficiently dense. Our proposed tests involve a combination of hard-thresholded nonlinear statistics of the adjacency matrix, the analysis of which may be of independent interest. In contrast with previous work, our non-asymptotic upper and lower bounds match for any configuration of $n_1, n_2, k_1, k_2$.

## 1   Introduction

The analysis of community structure is a fundamental task in statistical network science [27, 24]. In the 2000's, a series of striking works including (but not limited to) [3, 9, 30, 53, 56] observe the ubiquity of community-based organization in real world network data. Inspired by these observations, a vast literature has emerged over the past two decades attempting to understand the statistical limits of detecting and recovering planted communities in random graphs [20, 48, 50, 59, 4, 8, 49], detecting geometry in graphs defined via latent metric spaces [14, 13, 44, 25], and establishing fundamental computational limits for the community detection and recovery problems [31, 16, 60, 46, 12]. Here, we draw a careful distinction between *community detection* and *community recovery*. In the community detection problem, also sometimes referred to as the Planted Dense Subgraph (PDS) problem, the statistician observes a random graph and aims to determine whether the graph was drawn from an Erdős-Renyi family, or if it contains a densely connected community of vertices (i.e., a dense subgraph). In contrast, the community recovery problem involves assigning each vertex in the observed graph to a latent community, under the assumption that such community structure exists. We focus on community detection in the present work, and we refer readers to [1] for a comprehensive review of the results pertaining to the recovery problem in the stochastic block model.

Existing work studying community detection can be divided into two categories. The first camp, including [4, 59, 57, 61], focuses on detecting the presence of a small community of $k$ nodes in a high-dimensional graph of $n$ nodes, typically in an asymptotic setting with $k = o(n)$. In these papers, the authors focus on deriving sufficient and necessary conditions on the difference in connection

---

[*]The authors contributed equally to this work and are ordered alphabetically.

39th Conference on Neural Information Processing Systems (NeurIPS 2025).

probabilities between the planted subgraph and the rest of the vertices for which detection of the subgraph is possible. While these works are able to give asymptotically precise characterizations of the detection boundary for their problems of interest, their results are limited in generality, as they constrain themselves to fully symmetric settings (the adjacency matrices of the full graph and the subgraph, with the exception of [57], are square) and make asymptotic assumptions that prohibits the appearance of complex phenomena; see for instance equation (6) in [4].

A parallel line of work considers the community detection problem in stochastic block models (SBM's) [38, 48, 51, 2, 6, 7, 8, 28]. Here the focus is on the statistician's ability detect the presence of two or more communities in the observed graph, and deriving conditions on the difference in expected degree between the communities under which detection is possible. The early work [51] gives fundamental results related to the famed Kesten-Stigum threshold, although they are confined to the restricted case of two equally sized communities. While recent papers such as [8, 52] allow for a growing number of potentially asymmetric communities, these works still required the average expected degree across communities to be held constant, which prevents their results from applying to graphs with highly imbalanced communities. The restriction is lifted in [29], in which the authors study community detection in the degree-corrected block model. However, similar to the works of [4, 59, 57], the results of [29] hold only in a particular asymptotic regime.

Given the limitations of existing work, there remains a great interest in studying community detection in random graph models beyond those covered by the literature. In particular, previous results do not apply to the family of *bipartite graphs*, which are defined as sets of edges between two disjoint sets of vertices. In practice, the two sets of vertices may be of very different size, meaning that the results tailored for square adjacency matrices [4, 59, 2, 51] do not capture the subtleties of community detection in bipartite graphs. Furthermore, any planted subgraph in a bipartite graph is, by definition, itself bipartite. While the recent paper [57] considers the detection of bipartite communities, they assume that these communities are planted in a larger non-bipartite graph and hence, similarly to [4, 59, 2, 51], fail to fully characterize the detection boundary in the bipartite case. We also comment that bipartite graphs can be thought of as a special case of a multi-community SBM; however, as outlined above, the prior work studying this problem introduces its own limitations [8, 52, 29]. As bipartite graphs are increasingly used as modeling tools in the diverse fields of biology and medicine [54], information science [47], and game theory [55], this lack of understanding in the bipartite setting constitutes a significant gap in the statistics and machine learning literature.

In this paper, we aim to address this gap. Following [4, 59], we formalize the problem of detecting a community in a random bipartite graph as a hypothesis testing problem. We observe a random bipartite graph $\mathbf{G}$ defined on disjoint sets of vertices (nodes) $\mathcal{V}_1$ and $\mathcal{V}_2$ of size $n_1$ and $n_2$ respectively. Under the null hypothesis, an edge appears between any two nodes in $\mathbf{G}$ with equal probability $p_0 > 0$. Under the alternative hypothesis, there exist subsets of nodes $\mathcal{V}_{K_1} \subset \mathcal{V}_1$ and $\mathcal{V}_{K_2} \subset \mathcal{V}_2$ of size $k_1$ and $k_2$ such that any pair of nodes in $\mathcal{V}_{K_1} \times \mathcal{V}_{K_2}$ are connected with an elevated probability no less than $p_0 + \delta$, where $\delta > 0$. We derive the *minimax rate of separation* $\delta^*$ [35], which is defined as the smallest value of $\delta$ such that consistent community detection is possible (see Equation (4) for a formal definition). Our results are non-asymptotic in nature, and hold for any values of $n_1, n_2, k_1$, and $k_2$. This degree of precision allows us to describe subtle phase transitions in the separation rate $\delta^*$ which have not been observed in the existing community detection literature.

The paper is structured as follows. In Sections 1.1, 1.2, and 1.3 we review relevant literature, outline our contributions, and collect notation used through the paper. In Section 2 we formally state our problem of interest. Our main results and a discussion thereof are given in Section 3 and Section 4. We conclude with a summary of the limitations of our results and promising directions for future work in Section 5. All of the proofs are provided in the supplement.

## 1.1 Prior work

The framework that we use to study optimal community detection in random graphs was pioneered by Arias-Castro and Verzelen [4, 59]. In [4], the authors provide matching asymptotic upper and lower bounds on the minimax risk of detecting a community of vanishing size in a dense graph. They also handle the cases when the size of the community and the baseline connection probability $p_0$ are unknown, and provide polynomial-time tests using a convex relaxation of the max degree test, following [11]. In [59], the authors extend their analysis to sparse graphs. While the papers [4, 59] broke significant ground, their results are still limited, as they only consider the restricted

fully symmetric setting of $n_1 = n_2$ and $k_1 = k_2$. Subsequent works derive analogous theoretical results for hypergraphs [62, 61], in which edges can be drawn between more than two vertices. The work [33] studies community detection under information constraints, namely that the statistician can only access small parts of the graph via non-adaptive edge queries. More recently, [57] considers the detection of a bipartite (i.e., imbalanced) community in a non-bipartite graph. We provide a detailed comparison of our results to the results of [57] in Section 3.

In the stochastic block model (SBM) literature, the community detection problem is posed as the task of distinguishing an SBM with a given set of parameters from an Erdős-Renyi graph. This problem was first considered by [38], whose results were later refined by [51, 48, 2], which established that detection is possible if and only if the signal strength is above the Kesten-Stigum threshold, as conjectured by [23]. In [6, 7], the authors prove analogous results for asymptotically growing expected degrees and provide optimal hypothesis tests based on signed cycle statistics. The work of [8] provides an impossibility result for distinguishability for SBM's with a possibly growing number of communities, and [28] analyzes a powerful test in this setting constructed using small subgraph statistics.

Under the framework of [4, 59], community detection is equivalent to detecting a submatrix in the observed adjacency matrix. The seminal work of [15] provides matching upper and lower bounds for detecting a sparse submatrix of elevated mean in a matrix of Gaussian random variables under a particular asymptotic regime. Following this, [46] initiates a rigorous study of statistical-computational gaps in the submatrix detection problem which was continued by [16, 32, 12]. In [45], the authors study the problem of detecting a planted sparse sub-tensor in a high dimensional Gaussian tensor. Finally, [22] derives upper and lower bounds for detecting the presence of multiple sparse submatrices in Gaussian noise that are tight up to log factors.

## 1.2 Our contributions

In this paper, we make the following contributions.

1. We fully settle the non-asymptotic expression of the minimax rate $\delta^*$ defined in (4), which holds provided the graph is sufficiently dense, as specified in Assumption 1. In contrast to the previous literature, our bounds always match up to multiplicative constants for any possible values of $n_1, n_2, k_1$ and $k_2$, especially in the under-explored unbalanced regimes where $k_1 \ll k_2$ or $n_1 \ll n_2$. Our results reveal subtle phase transitions that, to the best of our knowledge, had not been documented in previous work.

2. Our lower bound on $\delta^*$ stated in Theorem 1 holds without requiring Assumption 1 and uses a very precise application of the second moment method [42]. It requires several non-trivial lemmas for controlling the moment-generating function of the product of binomial random variables, and substantially departs from the lower bound strategies proposed in [15] and [4].

3. Our upper bound stated in Theorem 2 is achieved by carefully combining three tests: A standard *total degree* test and two entirely novel tests referred to as the *truncated degree* test and the *max truncated degree* test. The two novel tests are defined using truncated non-linear functions of the adjacency matrix, building on the truncated $\chi^2$ test recently studied in the Gaussian sequence model [21, 40, 43, 19]. However, substantial modifications of the truncated $\chi^2$ test are needed to address two challenges:

   (a) Moving from the vector case to the matrix case, which requires subtle Bonferroni corrections of the classical truncated $\chi^2$ test.

   (b) Handling matrices whose entries are Bernoulli rather than Gaussian random variables, which require significant adjustments of the truncated $\chi^2$ test to carefully manage the complex sub-poissonian tails of the binomial distribution.

   Our upper bound holds under Assumption 1, which we discuss thoroughly in later sections. We also prove a series of lemmas in the supplement that we use to control the Type I and Type II errors of these tests, and we anticipate that these results will be of interest to researchers studying related problems.

## 1.3 Notation

The following notation will be used throughout the paper. For $p \in \mathbb{N}$, let $[p] := 1, ..., p$. We use $\mathcal{P}_k(n)$ to denote the set of subsets of $[n]$ of size $k$. For $a, b \in \mathbb{R}$, denote $a \vee b := \max\{a, b\}$ and $a \wedge b =: \min\{a, b\}$. We will use $a \lesssim b$ if there exists a constant $C > 0$ depending on $\eta$ such that $a \leq Cb$. We say $a \asymp b$ if $a \lesssim b$ and $b \lesssim a$. For two sets $A_1$ and $A_2$, we denote $A_1 \times A_2 = \{(i, j) : i \in A_1, j \in A_2\}$ as the Cartesian product of $A_1$ and $A_2$. For a finite set $A$, we use $|A|$ to denote the cardinality of $A$. We use $\mathbf{1}_{\{\cdot\}}$ as the indicator function, meaning that $\mathbf{1}_A = 1$ if the event $A$ occurs and $\mathbf{1}_A = 0$ otherwise. For a matrix $\mathbf{X}$, we use $X_{ij}$ to denote its $(i, j)_{\text{th}}$ entry. Given two probability distributions $\mathbb{P}$ and $\mathbb{Q}$, we use $\text{TV}(\mathbb{P}, \mathbb{Q}) = \sup_A |\mathbb{P}(A) - \mathbb{Q}(A)|$ to denote the total variation distance between $\mathbb{P}$ and $\mathbb{Q}$. We use $c, C, \bar{C}, C', c_1, c_2, c_3$, and $c_4$ to denote constants whose value may change between instances of their usage in the paper. We will also use the convention that $[a, b] = \emptyset$ for any two real numbers $a, b$ such that $a > b$. For any two integers $n, k \in \mathbb{N} \cup \{0\}$, we denote by $\binom{n}{k}$ the binomial coefficient, equal to $\frac{n!}{k!(n-k)!}$.

## 2 Problem statement

Let $n_1, n_2, k_1, k_2 \in \mathbb{N}$ be integers considered as fixed throughout the paper, and assume $k_1 \leq n_1$ and $k_2 \leq n_2$. Let $\mathbf{G} = (\mathcal{E}, \mathcal{V}_1, \mathcal{V}_2)$ be a random undirected bipartite graph defined on disjoint sets of vertices $\mathcal{V}_1 = \{V_{1,1}, \ldots, V_{1,n_1}\}$ and $\mathcal{V}_2 = \{V_{2,1}, \ldots, V_{2,n_2}\}$. We observe the adjacency matrix $\mathbf{A} \in \mathbb{R}^{n_1 \times n_2}$ with entries $A_{ij} = \mathbf{1}\{V_{1,i} \text{ is connected to } V_{2,j} \text{ by an edge}\}$ and assume that the edges are mutually independent Bernoulli random variables. For some parameter $p_0 \in [0, 1]$ assumed to be fixed throughout the paper, our objective is to determine whether each pair of vertices in $\mathcal{V}_1 \times \mathcal{V}_2$ is connected with probability $p_0$, or if there exists a community (i.e., a bipartite subset of vertices) that is connected with greater probability $p_1 \geq p_0 + \delta$ for some $\delta > 0$.

Formally, we formulate this testing problem in terms of the mean structure of the adjacency matrix $\mathbf{A}$. Define the matrix $\mathbf{P} = \mathbb{E}[\mathbf{A}] \in \mathbb{R}^{n_1 \times n_2}$, which represents the matrix of connection probabilities of the random graph $\mathbf{G}$. For any two subsets $K_1 \in \mathcal{P}_{k_1}(n_1)$, $K_2 \in \mathcal{P}_{k_2}(n_2)$ and any $\delta \geq 0$, we define

$$\Theta(K_1, K_2, \delta) = \left\{ \mathbf{P} \in \mathbb{R}^{n_1 \times n_2} \text{ s.t. } \begin{cases} \forall (i, j) \in K_1 \times K_2 : P_{ij} \geq p_0 + \delta \\ \forall (i, j) \notin K_1 \times K_2 : P_{ij} = p_0 \end{cases} \right\}.$$

A mean matrix $\mathbf{P}$ belongs to $\Theta(K_1, K_2, \delta)$ if the bipartite community indexed by $K_1 \times K_2$ is more densely connected than the rest of the graph. Our focus in this paper is to optimally detect an unknown bipartite community of size $k_1 \times k_2$. To this end, for any $\delta > 0$, we let

$$\Theta(k_1, k_2, n_1, n_2, \delta) = \bigcup_{K_1 \in \mathcal{P}_{k_1}(n_1)} \bigcup_{K_2 \in \mathcal{P}_{k_2}(n_2)} \Theta(K_1, K_2, \delta).$$

We consider the following testing problem

$$\text{H}_0 : \forall (i, j) \in [n_1] \times [n_2], P_{ij} = p_0 \quad \text{against} \quad \text{H}_1 : \mathbf{P} \in \Theta(k_1, k_2, n_1, n_2, \delta). \tag{1}$$

The hypothesis $\text{H}_0$ is equivalent to observing a bipartite Erdős-Renyi random graph with parameter $p_0$. The hypothesis $H_1$ is equivalent to the existence of an unknown bipartite community of size $k_1 \times k_2$ with connection probabilities at least $p_0 + \delta$.

A *test* is a measurable function of the observed data $\mathbf{A}$ taking its values in $\{0, 1\}$. We measure the quality of a test $\Delta$ by its *risk*, defined as the sum of its Type I and worst-case Type II errors. Specifically, denoting by $\mathbb{P}_{\mathbf{P}}$ the probability distribution of $\mathbf{A}$ for $\mathbf{P} \in \Theta(k_1, k_2, n_1, n_2, \delta)$, and letting $\mathbb{P}_0$ denote the distribution of $\mathbf{A}$ under the null hypothesis, the risk of a test $\Delta$ is defined as

$$\mathcal{R}(\Delta, \delta) = \mathbb{P}_0(\Delta = 1) + \sup_{\mathbf{P} \in \Theta(n_1, n_2, k_1, k_2, \delta)} \mathbb{P}_{\mathbf{P}}(\Delta = 0). \tag{2}$$

The *minimax risk* associated with the problem (1) is defined as

$$\mathcal{R}^*(k_1, k_2, n_1, n_2, \delta) = \inf_{\Delta} \mathcal{R}(\Delta, \delta), \tag{3}$$

where the infimum is taken over all tests $\Delta$. For a desired level of risk $\eta \in (0, 1)$, considered as a fixed constant throughout, we are interested in the *minimax separation rate* $\delta^*$ defined as

$$\delta^* = \inf \left\{ \delta > 0 : \mathcal{R}^*(k_1, k_2, n_1, n_2, \delta) \leq \eta \right\}. \tag{4}$$

The minimax separation rate $\delta^*$ encodes the difficulty of the testing problem. It is the infimal signal strength ensuring the existence of a test with Type I plus Type II errors controlled by the desired level of risk $\eta$. The goal of the paper is to derive the value of $\delta^*$ in terms of $n_1, n_2, k_1, k_2$ and $p_0$ up to absolute multiplicative constants, and to construct the minimax-optimal tests achieving a risk at most $\eta$ for the testing problem (1) when the separation satisfies $\delta \geq C\delta^*$ for some constant $C > 0$ depending only on $\eta$.

# 3  Main results

For any $k_1, k_2, n_1, n_2 \in \mathbb{N}$ and for a constant $C > 0$ to be chosen later, we define

$$\psi(k_1, k_2, n_1, n_2) = \frac{1}{k_1} \log\left(1 + \frac{n_2}{k_2^2} \log\left(e\binom{n_1}{k_1}\right)\right) \tag{5}$$

$$\phi(k_1, k_2, n_1, n_2) = \begin{cases} \frac{n_1}{k_1^2} \log\left(1 + \frac{n_2}{k_2^2}\right) & \text{if } \frac{n_1}{k_1^2} \leq C \\ \infty & \text{otherwise.} \end{cases} \tag{6}$$

$$\nu(k_1, k_2, n_1, n_2) = \frac{1}{k_1} \log\left(\frac{n_2}{k_2}\right) \mathbf{1}_{\left\{\frac{n_1 k_2}{k_1^2} \log\left(\frac{n_2}{k_2}\right) > 1\right\}} \tag{7}$$

To alleviate the notation, we will write

$$\phi_{12} = \phi(k_1, k_2, n_1, n_2) \qquad\qquad \phi_{21} = \phi(k_2, k_1, n_2, n_1)$$
$$\psi_{12} = \psi(k_1, k_2, n_1, n_2) \qquad\qquad \psi_{21} = \psi(k_2, k_1, n_2, n_1)$$
$$\nu_{12} = \nu(k_1, k_2, n_1, n_2) \qquad\qquad \nu_{21} = \nu(k_2, k_1, n_2, n_1).$$

Finally, we define the quantities

$$R := R(k_1, k_2, n_1, n_2) = \left(\psi_{12} + \psi_{21}\right) \wedge \phi_{12} \wedge \phi_{21} \tag{8}$$

and

$$\tilde{R} = \left(\psi_{12} + \nu_{21}\right) \wedge \left(\psi_{21} + \nu_{12}\right) \wedge \phi_{12} \wedge \phi_{21}.$$

We will prove that the minimax separation rate $\delta^*$ satisfies

$$(\delta^*)^2 \asymp p_0(1 - p_0)R,$$

for any values of $k_1 k_2, n_1$ and $n_2$, under Assumption 1 on $p_0$ given in Section 3.2.

## 3.1  Lower bound

The following theorem gives a lower bound on $\delta^*$ that holds for any values of $n_1, n_2, k_1, k_2$ and $p_0$.

**Theorem 1** *Let $\eta \in [0, 1]$ be given. There exist constants $\underline{c}, \bar{C} > 0$ that depend on $\eta$ only, such that if $\delta^2 \leq \underline{c} p_0(1 - p_0)R$ with (6) defined with $C = \bar{C}$, then it holds*

$$\mathcal{R}^*(k_1, k_2, n_1, n_2, \delta) > \eta.$$

Theorem 1 yields the lower bound $(\delta^*)^2 \geq c_\delta p_0(1-p_0)R$ by definition of $\delta^*$. The proof of Theorem 1 uses a careful application of the second moment argument [42, 37, 34, 36, 15, 4]. We refer the reader to the supplement for details.

## 3.2  Upper bound

Now we present a matching upper bound on the minimax rate of separation $\delta^*$. We do so by carefully combining three testing procedures, which are each constructed from the observed adjacency matrix $\mathbf{A}$. Our three hypothesis tests are defined as follows.

1. *Total degree test:* Let $\sigma_{\text{deg}} = \sqrt{n_1 n_2 p_0(1 - p_0)}$ and define the test statistic

$$t_{\text{deg}} = \frac{1}{\sigma_{\text{deg}}} \sum_{i=1}^{n_1} \sum_{j=1}^{n_2} (A_{ij} - p_0) \tag{9}$$

The total degree test is defined as $\Delta_{\text{deg}}^h = \mathbf{1}(t_{\text{deg}} > h)$ for a choice of threshold $h > 0$.

2. *Truncated degree test:* To define this test, we introduce the Bennett function, defined as

$$h_B(x) = (1 + x) \log(1 + x) - x, \qquad \forall x > -1. \tag{10}$$
$$h_B(-1) = 1. \tag{11}$$

For any $j \in [n_2]$, and $a \geq 1$, let $\bar{\sigma} = \sqrt{n_1 p_0 (1 - p_0)}$ and define

$$\bar{A}_j = \frac{1}{\bar{\sigma}} \sum_{i=1}^{n_1} (A_{ij} - p_0)$$

$$W_j = n_1 (1 - p_0) h_B \left( -\frac{\bar{\sigma} \bar{A}_j}{n_1 (1 - p_0)} \right) + n_1 p_0 h_B \left( \frac{\bar{\sigma} \bar{A}_j}{n_1 p_0} \right). \tag{12}$$

Letting $\nu_a^{n_1} = \mathbb{E}_0 \left[ W_1 | \bar{A}_1 \geq a \right]$ and $\tau > 0$ denote a parameter to be chosen later, define

$$t_{\text{trunc-deg},1} = \sum_{j=1}^{n_2} \left( W_j - \nu_\tau^{n_1} \right) \mathbf{1}(\bar{A}_j \geq \tau). \tag{13}$$

The truncated degree test is as $\Delta_{\text{trunc-deg},1}^h = \mathbf{1}(t_{\text{trunc-deg},1} > h)$ for some threshold $h > 0$.

3. *Max truncated degree test:* For any $J_1 \in \mathcal{P}_{k_1}(n_1)$ and any $j \in [n_2]$, we let $\sigma_{J_1,j} = \sqrt{k_1 p_0 (1 - p_0)}$ and define

$$t_{J_1,j} = \frac{1}{\sigma_{J_1,j}} \sum_{i \in J_1} (A_{ij} - p_0)$$

$$W_{J_1,j} = k_1 (1 - p_0) h_B \left( -\frac{\sigma_{J_1,j} t_{J_1,j}}{k_1 (1 - p_0)} \right) + k_1 p_0 h_B \left( \frac{\sigma_{J_1,j} t_{J_1,j}}{k_1 p_0} \right). \tag{14}$$

Letting $\nu_a^{k_1} = \mathbb{E}_0 \left[ W_{J_1} | t_{J_1,j} \geq a \right]$ and $\tau > 0$ be a parameter to be chosen later, define

$$t_{\text{max-trunc-deg},1} = \max \left\{ \sum_{j=1}^{n_2} \left( W_{J_1,j} - \nu_\tau^{k_1} \right) \mathbf{1}(t_{J_1,j} > \tau) \, \middle| \, J_1 \in \mathcal{P}_{k_1}(n_1) \right\}. \tag{15}$$

Finally, for a choice of threshold $h > 0$, we define the maximum truncated degree test as $\Delta_{\text{max-trunc-deg},1}^h = \mathbf{1}(t_{\text{max-trunc-deg},1} > h)$.

Analogously, we define the tests $\Delta_{\text{max-trunc-deg},2}^h$ and $\Delta_{\text{trunc-deg},2}^h$ by swapping the roles of $k_1, k_2$, and $n_1, n_2$ in the definitions of $\Delta_{\text{max-trunc-deg},1}^h$ and $\Delta_{\text{trunc-deg},1}^h$ respectively.

For some constant $c_1 > 0$ depending only on the desired level of risk $\eta$, we further combine the degree and truncated degree tests as follows. Let

$$\Delta_a^{h_1,h_2} = \begin{cases} \Delta_{\text{trunc-deg},1}^{h_1} & \text{if } \frac{n_2}{k_2^2} \geq c_1, \\ \Delta_{\text{deg}}^{h_2} & \text{otherwise} \end{cases} \quad \text{and} \quad \Delta_b^{h_1',h_2'} = \begin{cases} \Delta_{\text{trunc-deg},2}^{h_1'} & \text{if } \frac{n_1}{k_1^2} \geq c_1, \\ \Delta_{\text{deg}}^{h_2'} & \text{otherwise.} \end{cases}$$

Using these procedures as our building blocks, we construct our final test $\Delta^*$ by applying the relevant test depending on what part dominates in the expression of the rate $R$. Specifically, our optimal test is defined as

$$\Delta^* = \begin{cases} \Delta_{\text{max-trunc-deg},1}^{h_3} & \text{if } \tilde{R} = \psi_{12} + \nu_{21} \\ \Delta_{\text{max-trunc-deg},2}^{h_3} & \text{if } \tilde{R} = \psi_{21} + \nu_{12} \\ \Delta_a^{h_1,h_2} & \text{if } \tilde{R} = \phi_{12} \\ \Delta_b^{h_1',h_2'} & \text{if } \tilde{R} = \phi_{21}. \end{cases}$$

Our upper bound on the risk of $\Delta^*$ holds under the following assumption, which guarantees that the observed graph $\mathbf{G}$ is sufficiently dense.

**Assumption 1 (Graph density.)** *There exists a constant $C_\eta > 0$ depending only on $\eta$ such that the baseline connection probability $p_0$ satisfies $p_0 \leq 1/4$ and*

$$
p_0 \geq \begin{cases}
\frac{C_\eta}{k_1 k_2} \log\left(e \binom{n_1}{k_1}\binom{n_2}{k_2}\right) & \text{if } \tilde{R} = (\psi_{12} + \nu_{21}) \wedge (\psi_{21} + \nu_{12}) \\
\frac{C_\eta}{n_1} \log\left(1 + \frac{n_2}{k_2^2}\right) & \text{if } \tilde{R} = \phi_{12} \text{ and } n_2 > k_2^2 \\
\frac{C_\eta}{n_2} \log\left(1 + \frac{n_1}{k_1^2}\right) & \text{if } \tilde{R} = \phi_{21} \text{ and } n_1 > k_1^2 \\
\frac{C_\eta}{n_1 n_2} & \text{otherwise.}
\end{cases}
$$

The upper bound $p_0 \leq \frac{1}{4}$ in Assumption 1 is used to apply Slud's inequality for binomial anti-concentration [58]. We invoke the different cases of Assumption 1 depending on the exact construction of $\Delta^*$. Assumption 1 places us in a quasi-normal moderate deviation regime, which enables us to control the tails of the test statistics (9), (13), and (15) with sufficient precision. These tests are based on binomial statistics, which exhibit sub-Gaussian concentration properties in the moderate deviation regime and sub-poissonian concentration in the large deviation regimes (see, e.g., [5, 17, 39, 18]). This dichotomy has also been noted in the two companion papers [4] and [59], which split the analysis between the case of *dense* graphs, where a quasi-normal regime emerges (see the first row of Table 1 in [4]), and *sparse* graphs, where fundamentally different behaviors occur due to lower connectivity. In this paper, we follow the same approach by restricting to dense graphs as specified in Assumption 1, and leave the case of sparse graphs for future work. Furthermore, Assumption 1 has a natural probabilistic interpretation, as codified in the following propositions.

**Proposition 1** *Let $c > 0$. There exists a constant $C > 0$ such that if*

$$
\frac{C}{k_1 k_2} \log\left(e \binom{n_1}{k_1}\binom{n_2}{k_2}\right) \leq p_0 \leq \frac{1}{4},
$$

*then it holds $\mathbb{P}_0 \left(\mathbf{G} \text{ has an empty } k_1 \times k_2 \text{ bipartite subgraph}\right) \leq c$.*

*Here, we say that a subgraph $\mathbf{G}_1$ of $\mathbf{G}$ is empty if all the vertices in $\mathbf{G}_1$ has degree equal to 0.*

**Proposition 2** *Let $c > 0$. There exists a constant $C > 0$ such that if*

$$
\frac{C}{n_2 k_1} \log\left(e \binom{n_1}{k_1}\right) \leq p_0 \leq \frac{1}{4},
$$

*then it holds $\mathbb{P}_0 \left(\exists I \in \mathcal{P}_{k_1}(n_1) : \sum_{i \in I} \sum_{j=1}^{n_2} A_{ij} = 0\right) \leq c$.*

Propositions 1 and 2 shows that Assumption 1 prevents the occurrence of extreme events under $\mathbb{P}_0$ that would otherwise inflate the Type I error of our testing procedures, especially the max truncated degree test in (15). We remark that the lower bound on $p_0$ assumed in Proposition 2 is stronger than what we need in Assumption 1; we include this result to provide intuition.

The following theorem guarantees that the minimax risk of $\Delta^*$ is small whenever $\delta$ is large enough.

**Theorem 2** *Let $\eta \in [0, 1]$ be given. Suppose that Assumption 1 holds. Then there exist constants $C_\delta, \bar{C} > 0$ and thresholds $h_1, h_2, h_3, h_4 > 0$ such that if $\delta^2 \geq C_\delta p_0 (1 - p_0) R$ with (6) defined with $C = \bar{C}$, then the test $\Delta^*$ satisfies*

$$
\mathcal{R}(\Delta^*, \delta) < \eta.
$$

Theorem 2 demonstrates that our proposed test $\Delta^*$ is able to match the lower bound on $\delta^*$ given in Theorem 1 under Assumption 1. Combining Theorems 1 and 2, we have therefore identified the minimax rate of separation $\delta^*$ up to constants.

## 4 Discussion

### 4.1 Discussion on the proposed tests

Our test $\Delta^*$ is a careful combination of three testing procedures. The total degree test has been commonly applied in previous works [4, 57], along with the *scan test*, which consists of scanning

over all possible subgraphs of size $k_1 \times k_2$ and rejecting if one of them contains an unusually large number of edges. Notably, our results do not make use of the scan test, and highlight that it can always be successfully replaced by a max truncated degree test, with lower time complexity[2].

Our two novel tests $\Delta_{\text{trunc-deg},1}^{h}$ and $\Delta_{\text{max-trunc-deg},1}^{h_3}$ build on the truncated $\chi^2$ test recently developed in the Gaussian sequence model. Given a vector $X \in \mathbb{R}^d$, the truncated $\chi^2$ test statistic is given by

$$T = \sum_{j=1}^{d} (X_j^2 - \nu_a)\mathbf{1}\big(|X_j| \geq a\big)$$

for some $a > 0$ and where $\nu_a = \mathbb{E}\big(Z^2\big||Z| \geq a\big)$. On top of being fast to compute, this test has been shown to be optimal in the Gaussian sequence model for detecting $s$-sparse alternatives separated in $\ell_2$ norm when $s \leq \sqrt{d}$ [21], whereas the max test, which rejects for large enough values of $\max\big\{\sum_{j\in J} X_j^2 | J \in \mathcal{P}_s(d)\big\}$, is known to be statistically suboptimal [10] while exhibiting exponential time complexity.

We aim to adapt this test to the matrix case where $X \in \mathbb{R}^{d_1 \times d_2}$. For the sake of discussion, we will focus on the truncated degree test (13), the ideas underlying the max truncated degree test (15) being analogous. An idea for adapting the truncated $\chi^2$ test would be to use

$$\sum_{i=1}^{n_1} \Big(\Big(\sum_{j=1}^{n_2} X_{ij}\Big)^2 - \nu_a\Big)\mathbf{1}\big(\sum_{j=1}^{n_2} X_{ij} \geq a\big)$$

for some suitable re-centering parameters $\nu_a$ and $\nu_a'$. Whereas squares of Gaussian random variables concentrate sub-exponentially (see Lemma 5 in [43]), squares of binomial distributions do not exhibit such favorable concentration properties, due to their sub-poissonian tails. In the definition of the test statistics (12) and (14), we therefore replace the square function with a carefully chosen function with slower growth. Focusing on the truncated degree test for illustration, and recalling (12), we can check that

$$W_j = n_1(1-p_0)\,h_B\Big(-\frac{\bar{\sigma}\bar{A}_j}{n_1(1-p_0)}\Big) + n_1 p_0\,h_B\Big(\frac{\bar{\sigma}\bar{A}_j}{n_1 p_0}\Big) \asymp \bar{\sigma}\bar{A}_j \log(1 + \frac{\bar{A}_j}{\bar{\sigma}}).$$

When $\bar{A}_j \leq \bar{\sigma}$, which exactly corresponds to the moderate deviation regime for binomial distributions, we obtain $W_j \asymp \bar{A}_j^2$ and our test statistic $t_{\text{trunc-deg},1} = \sum_{j=1}^{n_2}\big(W_j - \nu_\tau^{n_1}\big)\mathbf{1}(\bar{A}_j \geq \tau)$ becomes analogous to the truncated $\chi^2$ test described above. In the large deviation regime where $\bar{A}_j \gg \bar{\sigma}$, we have $W_j \asymp \bar{\sigma}\bar{A}_j \log(\frac{\bar{A}_j}{\bar{\sigma}})$, which grows less fast than $\bar{A}_j^2$ and mitigates the extreme values of the binomial distribution.

Heuristically, the truncated degree test is optimal when one sub-graph of the elevated community is large and the other is small. The max-truncated degree test is optimal when both sub-graphs are small. To demonstrate this: Suppose that one of the communities is very large, for example $k_2 = n_2$ to fix ideas, and the other one is small ($k_1 \ll n_1$). In the corresponding adjacency matrix, all entries within a row are i.i.d. Bernoulli random variables with the same probability parameter. It is well-known that a sufficient statistic of $m$ i.i.d. data points $X_1, \ldots, X_m \sim \text{Bernoulli}(p)$ is the sum $\sum_{i=1}^{m} X_i \sim \text{Binomial}(m, p)$. Therefore, the column vector obtained as the row-wise sum of the adjacency matrix is a sufficient statistic of the data. This vector is simply the vector of node degrees in the first sub-graph of the bipartite graph. Therefore, when one of the two sub-graphs is large enough, the problem can be reduced to a vector-based problem. A classical test statistic in such cases is the truncated chi-square test, which we apply to the degree vector with a slight refinement to account for the sub-Poissonian tails of the binomial distribution. This yields the truncated degree test.

When both sub-graphs of the elevated community are small, collapsing the matrix into a single vector (by summing over rows or columns) discards too much information from the data. In this case, a natural idea would rather be to scan over all possible bipartite subgraphs of size $k_1 \times k_2$ and reject the null hypothesis when one such subgraph contains an unusually large number of edges–an approach proposed in [57]. Unfortunately, this procedure is not optimal. Instead, we propose a non-trivial refinement that builds on the truncated chi-square test, adapted to tackle two key challenges: 1) the

---

[2]Computing $\Delta_{\text{max-trunc-deg},1}^{h_3}$ requires $O(k_1 n_2 \binom{n_1}{k_1})$ operations, rather than $O(k_1 k_2 \binom{n_1}{k_1}\binom{n_2}{k_2})$ for the scan test.

data are matrix-valued, not a vector-valued, and 2) the entries are Bernoulli rather than Gaussian random variables. Our procedure, the max-truncated degree test as formally defined in equation (14), is obtained by scanning and summing over subsets along one dimension of the matrix, applying a non-linear transformation based on the Bennett function, and truncating along the other dimension – analogous to the truncated chi-square test statistic. This procedure captures subtle concentration effects of bipartite graphs when both communities are small and achieves the minimax optimal risk in this regime.

## 4.2 Comparison with existing results

Through Theorems 1 and 2, we have shown

$$(\delta^*)^2 \asymp p_0(1 - p_0)R, \tag{16}$$

with $R$ defined in (8). We can understand the nuances of (16) by first considering our results when restricted to the fully balanced (effectively non-bipartite) setting. Suppose that $k_1 = k_2 = k$ and $n_1 = n_2 = n$, and that $n \leq Ck^2$. From the definition of $R$, we can rewrite (16) as

$$\frac{(\delta^*)^2}{p_0(1 - p_0)} \asymp \frac{1}{k} \log\left(\frac{n}{k}\right) \wedge \frac{n}{k^2} \log\left(1 + \frac{n}{k^2}\right). \tag{17}$$

If $\frac{n}{k^2} \gg 1$ or $\frac{n}{k^2} \ll 1$, the form of $\delta^*$ in (17) matches that given by Theorem 2 of [4] in the moderate deviation regime imposed by Assumption 1. Our result is, in fact, more refined than that of [4], as we are able to precisely capture the phase transition around $\frac{n}{k^2} \asymp 1$ thanks to our truncated degree test.

The detection boundary for imbalanced bipartite graphs is more complex. The following proposition describes a new phase transition that emerges in the imbalanced bipartite setting, but not in the non-bipartite setting.

**Proposition 3** *Suppose that $k_1^2 \geq \bar{c}n_1k_2$ for a constant $\bar{c} > 0$ and $k_j \leq c_j n_j$ for $j \in \{1, 2\}$ where $c_1, c_2 > 0$ are sufficiently small constants. Additionally, suppose that there exists a constant $\alpha > 0$ such that $n_2 \geq k_2^{2+\alpha}$ and that $\frac{n_1}{k_1} \geq e \log(\frac{n_2}{k_2})$. Then it holds*

$$\frac{(\delta^*)^2}{p_0(1 - p_0)} \asymp \frac{1}{k_2} \log\left(1 + \frac{n_1 k_2}{k_1^2} \log(n_2)\right). \tag{18}$$

*In particular, this reveals a phase transition at $\frac{n_1 k_2}{k_1^2} \log(n_2) \asymp 1$.*

To our knowledge, (18) is described nowhere in the existing community detection literature, as previous works either impose restrictions on the shape of the observed graph and the community of interest (i.e. enforcing $n_1 = n_2$ and/or $k_1 = k_2$) or provide results that are not sharp for all configurations of $n_1, n_2, k_1$, and $k_2$. The recent paper of [57] presents the results that are most similar to ours in the literature. They consider the case $n_1 = n_2 = n$ with $k_1 \neq k_2$ in the asymptotic regime $k_1 + k_2 = o(n)$. We again emphasize that our results hold for $n_1 \neq n_2$ and any $k_1$ and $k_2$, and as such are immediately more general than those of [57]. However, we argue that our results are more precise even in the restricted case of $n_1 = n_2 = n$ and $k_1 + k_2 = o(n)$. Under the assumption that $p_0, \delta^* = \Theta(n^{-\alpha})$ for $\alpha \in (0, 2]$, in Theorem 1 of [57] the authors show that

$$\frac{(\delta^*)^2}{p_0(1 - p_0)} = \Omega\left(\frac{1}{k_1 \wedge k_2} \wedge \frac{n^2}{k_1^2 k_2^2} \wedge \frac{n}{k_1^2 \vee k_2^2}\right),$$

and in Theorem 2, they show

$$\frac{(\delta^*)^2}{p_0(1 - p_0)} = O\left(\frac{\log(n)}{k_1 \wedge k_2} \wedge \frac{n^2}{k_1^2 k_2^2} \wedge \frac{n \log(n)}{k_1^2 \vee k_2^2}\right).$$

We refer readers to equations (14) and (15) in [57] and the surrounding discussion for details. Their lower bound on $\delta^*$ is loose by a logarithmic factor in $n$, and the asymptotic assumption $\delta^* = \Theta(n^{-\alpha})$ excludes the regime in which the complex rate that we derive in (16) emerges. As such, the results of [57] do not describe, for instance, the phase transition (18). We also comment on the assumptions on $p_0$ made in [57] and in our work. Recall that the results of [57] require $p_0 \gtrsim n^{-\alpha}$ for $\alpha \in (0, 2]$, where under our Assumption 1, we require at least $p_0 \gtrsim \frac{1}{n^2}$ and, in certain regimes, we need

$p_0 \gtrsim \frac{1}{k_1 k_2} \log(\binom{n}{k_1}\binom{n}{k_2})$. This assumption is stronger than that of [57], and hence excludes our results from holding for all values of $p_0$ considered by [57]. However, as their results are not sharp, they are not able to describe the entirety of the detection boundary in the case when our Assumption 1 does not hold. We anticipate that the minimax rate of separation $\delta^*$ may actually differ non-trivially from the rate derived in [57] in this setting, and we view the extension of our results beyond Assumption 1 as a crucial piece of future work. Finally, we remark that [26] extends the results of [57] beyond planted bipartite subgraphs, but they do so in the same asymptotic setting and thus suffer from the same limitations.

## 5    Conclusion, limitations, and open problems

We have presented a rigorous study of the community detection problem in bipartite graphs. We present a lower bound on the minimax rate of separation, and describe a novel optimal testing strategy that achieves the minimax rate when the observed graph is dense. Our non-asymptotic results hold for all possible dimensions of the observed graph and the planted subgraph, which reveals new behavior in the detection boundary. Here, we outline some limitations and opportunities for future work.

**Adaptivity to unknown $p_0, k_1,$ and $k_2$.** All three of our tests require knowledge of $p_0$, the connection probability under the null hypothesis, which is typically unavailable in practice. In [4], the authors consider this problem in the case of non-bipartite graphs, and find that community detection without knowledge of $p_0$ exhibits substantially different behavior. We anticipate that similar phenomena will arise in the bipartite setting.

Furthermore, our max truncated degree test requires knowledge of the size of the community as encoded in $k_1$ and $k_2$. Optimal adaptivity to unknown $k_1$ and $k_2$ is achieved in previous works such as [15, 4] by scanning over possible values of $k_1$ and $k_2$ and performing a Bonferroni correction. However, this not an option for us, since the tails of the test statistic (15) are too heavy to still be optimal after a union bound. One possible strategy is to use Lepski-style adaption as in [41]; we leave this to future work.

**Beyond dense bipartite graphs.** The tightness of our upper bound in Theorem 2 relies on Assumption 1. Prior work in the non-bipartite community detection literature [59] suggests that the detection landscape is very different for sparse graphs. Understanding optimal detection in bipartite graphs when $p_0$ is extremely small will potentially require testing procedures and new techniques for proving lower bounds, likely based on the truncated second moment method [35, 15, 4, 59].

**Computational considerations.** Our max truncated degree test is unfeasible to compute on datasets of even moderate size. It is well established that the community detection problem in non-bipartite graphs exhibits a statistical-computational gap [46, 4, 31, 12, 22, 57, 26], meaning that it is not, in general, possible to optimally detect a planted community with a test that runs in polynomial time. We conjecture that this gap persists in the bipartite case, and establishing this phenomenon formally is an open problem of great interest.

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
