# OpenReview forum: "Optimal community detection in dense bipartite graphs"
_NeurIPS.cc/2025/Conference — NeurIPS 2025 poster_

### Official Review · Reviewer_1bpB · 2025-06-28

**Clarity:** 4
**Significance:** 3
**Originality:** 3
**Rating:** 5
**Confidence:** 4

**Summary:**

The authors consider the problem of detection between on the one hand: a bipartite Erdos-Rényi graph with bipartition sizes n_1, n_2 and probability of an edge p_0, and on the other hand a bipartite graph where the edge presence probability is, between two sets of respective sizes k_1 and k_2, elevated from p_0 to being larger than p_0+delta. Their objective is to determine, for a target risk level \eta (defined in Eq(2)), necessary and sufficient conditions on the magnitude of the elevation parameter delta, for the existence of a test Delta that discriminates between the two hypotheses with risk at most \eta.

They then provide a lower bound on the critical value of delta, given in theorem 1, below which no such test exists. Under additional Assumption 1 (which guarantees that the graph be sufficiently dense), they give in Theorem 2 an upper bound on the critical value of delta above which a sophisticated test succeeds. This upper bound is within a constant factor of the lower bound, so that under Assumption 1 the two theorems characterize the magnitude of the critical value of delta.

Finally the authors discuss implications of their results. They show that these generalize previously known thresholds in a balanced setting, and that their result reveals a novel phase transition specific to the unbalanced bipartite case (proposition 3).

A discussion of limitations and future directions concludes the paper.

**Questions:**

-At the bottom of page 3, \eta is first mentioned while not being defined yet, you may want to fiw that.

-Could you comment on the fact that the constant C discussed at the bottom of page 4 depends on \eta? What would happen in a regime where eta goes to zero?

-In statement of theorem 1, notation c_delta for the constant is a bit awkward, as it might lead the reader to believe that this constant does depend on delta. However this would not make sense. Please use an alternative notation.
-On page 6, line 106, under what probability distribution is expectation taken? P_0?

-In Eq(14), is the maximum over sets J_1?

-Can you comment on the choice of thresholds h_i used in Theorem 2?

**Ethical Concerns:**

["NO or VERY MINOR ethics concerns only"]

**Final Justification:**

After the discussion phase, I finally upgraded my rating to "accept": the paper is very solid, and even though it is of theoretical nature, its contribution is of interest to a significant subset of the NeurIPS comunity.

**Limitations:**

Yes

**Quality:**

3

**Strengths And Weaknesses:**

Strengths:The test setup considered is challenging, as the authors consider for alternative hypothesis a whole class of edge probabilities, specified by the elevation parameter delta. The result obtained is difficult, and it is interesting that it reveals a new phase transition for community detection specific to the unbalanced bipartite case. The underlying proof techniques seem powerful and delicate to use. The writing is good.

---

> ### Author Rebuttal · Authors · 2025-07-30
>
> 1. *At the bottom of page 3, \eta is first mentioned while not being defined yet, you may want to fix that.*
>
> Response: Thank you for pointing this out, we will make sure that $\eta$ is well-defined in the camera-ready version of our paper.
>
> 2. *Could you comment on the fact that the constant C discussed at the bottom of page 4 depends on \eta? What would happen in a regime where eta goes to zero?*
>
> Response: The constant $C$ will be increased as $\eta$ decreases. In the non-asymptotic minimax framework that we work in, $\eta$ is fixed between $0$ and $1$ prior to the subsequent analysis, and therefore we need not concern ourselves with $C$ tending to infinity. This framework is standard in the literature (see the references [10, 11, 15, 17, 18, 19, 21, 39, 43, 59], among others, in the current version of our paper). We do not allow for $\eta = 0$, as it is not possible in a non-asymptotic setting to guarantee that the type 1 and 2 errors are identically 0. The case where $\eta \to 0$ is certainly of interest, but does not fit within our framework, as our goal is to provide non-asymptotic statistical guarantees.
>
> 3. *In statement of theorem 1, notation $c_\delta$ for the constant is a bit awkward, as it might lead the reader to believe that this constant does depend on delta. However this would not make sense. Please use an alternative notation.*
>
> Response: Thank you for this note on notation. We agree with you and will replace $c_\delta$ with $\underline{c}$ in the camera-ready version of our paper.
>
> 4. *On page 6, line 106, under what probability distribution is expectation taken? P_0?*
>
> Response: Yes, this expectation is taken under $P_0$. We will add a subscript to the expectation notation to make this clear in the camera-ready version of our paper.
>
> 5. *In Eq(14), is the maximum over sets J_1?*
>
> Response: Yes, that is correct. The max truncated degree is constructed by scanning over subsets of size $k_1$ (or $k_2$) of the number of rows (or columns) and computing the average truncated degree statistics on each subset of indices. We will adjust the notation in the paper to improve clarity for the reader.
>
> 6. *Can you comment on the choice of thresholds h_i used in Theorem 2?*
>
> Response: We present precise choices of $h$ and $\tau$ that achieve the optimal testing risk for each of the truncated degree tests in Section B of the supplement. We refer the Referee to our response to Reviewer QZpt for the precise values of $h$ and $\tau$ that are used for each test. We will include these values in the camera ready version of our paper for clarity.

---

> > ### Comment · Reviewer_1bpB · 2025-08-06
> >
> > I would like to thank the authors for their thorough replies to my questions / suggestions.

---

### Official Review · Reviewer_Kxva · 2025-07-01

**Clarity:** 2
**Significance:** 2
**Originality:** 2
**Rating:** 4
**Confidence:** 2

**Summary:**

The authors present new statistical tests for community detection in random bipartite graphs and prove that they are minimax optimal.

**Questions:**

- *theorem 2 of [2]  deals with G(n,p,Q) where p_i’s is the probability of a node belonging to community i, and pi’s are in (0,1). This does not seem to be consistent  with the claim that they “are confined to the restricted case of two equally sized communities”. Please clarify.

- “the two sets of vertices may be of very different size, meaning that the results tailored for square adjacency matrices … do not capture the subtleties of community detection in bipartite graphs.” This seems to suggest that using a square adjacency matrix to represent a bipartite graph might be restrictive. Please clarify whether this is what you meant.

**Ethical Concerns:**

["NO or VERY MINOR ethics concerns only"]

**Final Justification:**

There are some inaccuracies in the paper, which however seem minor. Given that I am not familiar with this line of work I feel more comfortable to let the other reviewers decide whether accepting it or not.

**Limitations:**

yes

**Quality:**

2

**Strengths And Weaknesses:**

Strengths:
- the problem studied in the paper is interesting
- the paper is well written and organized

Weaknesses:
- a statistics conference or journal might be a better fit
- it is difficult to assess the novel contribution w.r.t. to the state of the art (see questions below)

---

> ### Author Rebuttal · Authors · 2025-07-30
>
> 1. *Theorem 2 of [2] deals with G(n,p,Q) where p_i’s is the probability of a node belonging to community i, and pi’s are in (0,1). This does not seem to be consistent with the claim that they “are confined to the restricted case of two equally sized communities”. Please clarify.*
>
> Response: We thank the reviewer for pointing out this inaccuracy. It is true that the work of [2] allows for multiple communities of possibly unequal size, and we will correct this in the literature review section of our paper. However, they are not studying the same problem as us. “Community detection” in their language refers to clustering each vertex in a non-bipartite SBM and recovering the true underlying clusters asymptotically. In contrast, we consider the problem of testing whether a community exists in a bipartite graph, in a fully non-asymptotic setting. We will correct this slight inaccuracy in our literature review. However, we emphasize that this does not detract from the novelty of our results, as no optimal rates were previously known for arbitrary values of $k_1, k_2, n_1$ and $n_2$ in our detection problem.
>
> 2. *“The two sets of vertices may be of very different size, meaning that the results tailored for square adjacency matrices … do not capture the subtleties of community detection in bipartite graphs.” This seems to suggest that using a square adjacency matrix to represent a bipartite graph might be restrictive. Please clarify whether this is what you meant.*
>
> Response: For bipartite graphs with two populations of $n_1$ and $n_2$ nodes respectively, the adjacency matrix is a rectangular matrix of size $n_1 \times n_2$. This contrasts with a non-bipartite graph with n nodes, whose adjacency matrix is a square matrix of size $n \times n$. Results developed in the case of non-bipartite graphs–that is, for square adjacency matrices–are therefore inapplicable to the case of bipartite graphs whose adjacency matrices can have substantially more rows than columns, or vice versa. Of course, it is always possible to represent a rectangular matrix as a square matrix by simply adding empty rows or columns. However, existing results for square adjacency matrices still do not apply in this case. Indeed, they assume that under the null hypothesis, all entries are iid Bernoulli random variables with non-zero parameter, which excludes cases where adjacency matrices have entire rows or columns that are identically zero (see e.g. Verzelen and Arias-Castro, Community detection in dense random networks, AoS 2024).
>
> It may be the case that the reviewer thinks that the setting concerns the detection of a bipartite community of size $k_1 \times k_2$ in a larger, non-bipartite graph of size $n \times n$. In reality we consider the detection of a bipartite community of size $k_1 \times k_2$ within a larger bipartite graph of size $n_1 \times n_2$.  The former setting is considered in (Rotenberg, Huleihel and Shayevitz, Planted Bipartite Graph Detection, IEEE TIT 2024). We present a thorough comparison of our results to theirs in Section 4 of our paper. To summarize, the restrictive non-bipartite setting of (Rotenberg, Huleihel and Shayevitz, Planted Bipartite Graph Detection, IEEE TIT 2024) prevents them from articulating the new phrase transitions with respect to relative sizes of the two disjoint sets of vertices that constitute the bipartite graph as we derive in this work, specifically in Proposition 3. This is one of the primary contributions of our paper. Furthermore, their results are loose by log factors, which we are able to tighten up to constants.
>
>
> 3. *Regarding novelty compared to the state of the art*
>
> The Referee requested clarification regarding our paper's contributions in relation to the existing literature. We are happy to outline them more clearly here and hope this will address the Referee’s main concern.
>
> - We consider the problem of detecting a bipartite community of size $k_1 \times k_2$ in a bipartite graph with two populations of size $n_1$ and $n_2$, respectively. We allow the parameters $k_1, k_2, n_1, n_2$ to assume arbitrary values--provided of course that $k_1 \leq n_1$ and $k_2 \leq n_2$. This very general setting goes far beyond the current literature, where the observed graph is most often non-bipartite, that is $n_1 = n_2$ and the community is non-bipartite as well, that is $k_1 = k_2$ (see e.g. [4], (Arias-Castro and Verzelen, *Community detection in sparse random networks*, AoS 2015)). A few works consider imbalanced settings, e.g. [15], which considers Gaussian data rather than Bernoulli matrices, but operates under the very restrictive setting $k_1 \log(n_1/k_1) \asymp k_2 \log(n_2 / k_2)$ and also provides asymptotic results. The most closely related paper that we are aware of is [57], which considers arbitrary $k_1$ and $k_2$ but still $n_1 = n_2$, and obtains upper and lower bounds that do not match in all cases. They also do not propose any new tests compared to previous literature, therefore only obtaining suboptimal results. To the best of our knowledge, our paper is the first work to fully settle the **non-asymptotic rates** of community detection under any values of $k_1, k_2, n_1$ and $n_2$ by obtaining **matching upper and lower bounds in any case**. Our analysis reveals subtle phase transitions that, to the best of our knowledge, had not been established in previous work.  This level of generality requires substantially **new proof techniques and ideas**, and **entirely novel testing procedures** that we outline below and believe will be of independent interest to the community.
>
> - Our lower bound on $\delta^*$ stated in Theorem 1 is entirely novel and uses a very precise application of the second moment method. It requires several non-trivial lemmas for controlling the moment-generating function of the product of binomial random variables, and substantially departs from the lower bound strategies proposed in [15], [4] or [57].
>
> - Our upper bound stated in Theorem 2 is achieved by carefully combining three tests: A standard *total degree* test and two entirely novel tests referred to as the *truncated degree* test and the *max truncated degree* test. The two novel tests are defined using truncated non-linear functions of the adjacency matrix. We emphasize that our tests are non-trivial refinements of the truncated $\chi^2$ test recently studied in the Gaussian sequence model [21, 40, 43, 19]. Indeed, substantial modifications of the truncated $\chi^2$ test are needed to address two key challenges:
>      - Extending from the vector setting to the matrix setting, which requires careful Bonferroni-type corrections to control the type I error.
>      - Handling matrices whose entries are Bernoulli rather than Gaussian random variables, which require significant adjustments of the truncated $\chi^2$ test to carefully manage the complex sub-poissonian tails of the binomial distribution.
>
>
> We hope this discussion will convince the Referee of the novelty and significance of our results. We are happy to further revise the introduction or conclusion to make these points more prominent if the Referee believes this would benefit the reader.

---

> > ### Comment · Reviewer_Kxva · 2025-08-05
> >
> > Thanks for clarifying. I agree that the inaccuracy I pointed out does not detract from the novelty of your results. I trust the authors to address this in their paper. Concerning my second comment, any graph can be represented by a square adjacency matrix, which might not be the best representation in case the input graph is bipartite. I suggest to make more precise claims about that. I decided to raise my score to a borderline accept.

---

### Official Review · Reviewer_QZpt · 2025-07-02

**Clarity:** 3
**Significance:** 3
**Originality:** 3
**Rating:** 4
**Confidence:** 3

**Summary:**

The authors presented a rigorous study of the community detection problem in bipartite graphs. In particular, the paper presented a lower bound on the minimax rate of separation, and described a novel optimal testing strategy that achieves the minimax rate when the observed graph is dense. Their non-asymptotic results hold for all possible dimensions of the observed graph and the planted subgraph, which reveals new behavior in the detection boundary.

**Questions:**

(1) Given the reliance of the upper bound on Assumption 1, what are the key technical obstacles preventing the extension of your optimal testing strategies to the sparse graph setting? Do you anticipate the minimax rate $\delta^*$ itself changing fundamentally under sparsity?

(2) For truncated degree tests, how to choose the thresholds $\tau,h$?

**Ethical Concerns:**

["NO or VERY MINOR ethics concerns only"]

**Limitations:**

Yes.

**Paper Formatting Concerns:**

No.

**Quality:**

3

**Strengths And Weaknesses:**

Strengths: The paper presents non-asymptotic, matching upper and lower bounds for $\delta^*$ valid for all parameter configurations ($n_1,n_2,k_1,k_2$). This universality is a major advance over prior work constrained to asymptotic regimes, square matrices and symmetric communities. Besides, the introduction of the truncated degree test and max truncated degree test is significant. These adapt the truncated $\mathcal{X}^2$ concept to the challenges of binomial data and the matrix setting, requiring careful function design and handling of moderate vs. large deviations.

Weaknesses: (1) The upper bound and optimality of the proposed tests crucially depend on Assumption 1, requiring the graph to be sufficiently dense. This excludes the practically important regime of sparse bipartite graphs. (2) The max truncated degree test requires scanning over all subsets of $\mathcal{V}_1$ or $\mathcal{V}_2$, making it computationally infeasible for moderate-sized graphs. This paper doesn't offer any computationally efficient alternative achieving the rate, even heuristically. (3) This paper lacks empirical validation of the derived rates or the performance of the proposed tests.

---

> ### Author Rebuttal · Authors · 2025-07-30
>
> 1. *Given the reliance of the upper bound on Assumption 1, what are the key technical obstacles preventing the extension of your optimal testing strategies to the sparse graph setting? Do you anticipate the minimax rate itself changing fundamentally under sparsity?*
>
> Response: In the analysis of our testing procedures, Assumption 1 allows us to control the tail probabilities of the test statistics in question with sub-Gaussian concentration results as well as moderate and large deviation bounds for the binomial distribution. This is necessary to extract the correct rates that match the lower bound that we present in Theorem 1. We do not expect this analysis to carry over to the sparse regime, as our upper bound proof fundamentally breaks if one of the assumptions of the form $a \in [C, c\sigma]$ in Lemmas 18, 19, 20, 21 is violated. Such assumptions are guaranteed by Assumption 1 in the proof of Theorem 2.
>
> Indeed, in the sparse graph regime, we expect the optimal testing strategies to dramatically change compared to those developed in the paper. This difference has been described in (Arias-Castro and Verzelen, Community detection in dense random networks, AoS 2014) and (Arias-Castro and Verzelen, Community detection in sparse random networks, AoS 2015), which study a special case of our setting (namely, the non-bipartite setting) by imposing the restriction $n_1 = n_2$ and $k_1 = k_2$. In the sparse case, their test statistics are based on a “broad scan test statistic”, the number of triangles, the size of the largest connected component or the largest subtree in the graph, leading to substantially different statistical rates. Certain challenging regimes are also left unsolved. In our case, we expect the optimal test statistics to depend on delicate subpoissonian properties of sparse bipartite graphs, which will constitute a crucial piece of future research.
>
> 2. *For truncated degree tests, how to choose the thresholds tau and h?*
>
> We present precise choices of $\tau$ and $h$ that achieve the optimal testing risk for each of the truncated degree tests in the supplement; see the results in Section B for details. The $h$’s are defined at the bottom of page 37 and top of page 38 and $\tau$ is defined in statements of Lemma 16 and 17. In particular, for the truncated degree test defined over the columns of the adjacency matrix we take $\tau = \sqrt{C\log(1 + n_2/k_2^2)}$ and $h = 9(\sqrt{n_2 \exp(−c′\log(1 +n_2/k_2^2) c} + c)$.  For the max truncated degree test defined over the columns of the adjacency matrix we take $\tau = \sqrt{C\log(1 + n_2/k_2^2\log{n_1 \choose k_1})}$ and $h = 9(\sqrt{n_2 \exp(−c′\log(1 +n_2/k_2^2) c\log{n_1 \choose k_1}} + c\log{n_1 \choose k_1})$. We simply flip the indices of $k_1, k_2, n_1$, and $n_2$ if the test is rather defined over the columns. Here, $c, c'$ and $C$ are positive constants whose value may depend on the level of desired testing risk $\varepsilon$. We will add the thresholds to the main text for clarity.
>
> 3. *On computational challenges:*
>
> Response: There do exist known statistical-computational gaps in the community detection problem. This phenomenon was described formally in the non-bipartite case in (Verzelen and Arias-Castro, Community detection in dense random networks, AoS 2014), as well as in the bipartite subgraph case in (Rotenberg, Huleihel and Shayevitz, Planted Bipartite Graph Detection, IEEE TIT 2024). Therefore, we also expect statistical-computational gaps to arise in our setting. However, our objective was to settle the statistical aspects of the community detection problem in dense bipartite graphs, as no precise, optimal rates were previously known under any configuration of $k_1, k_2, n_1$ and $n_2$. We therefore leave the derivation of computational gaps to future work.
>
>
> As a side note on computational complexity: we never use the scan test, which was the only previously known test achieving near-optimal statistical rates of detection in specific regimes (Rotenberg, Huleihel and Shayevitz, Planted Bipartite Graph Detection, IEEE TIT 2024). Instead, we always replace it with our max-truncated-degree test. Although computing our test statistic requires exponential time, its time complexity still substantially improves upon that of the scan test, while also providing a statistically more powerful test in most regimes (see Section 4.1 for details).

---

> > ### Comment · Reviewer_QZpt · 2025-08-06
> >
> > Thank you for the clarification. The reviewer has no further comments or suggestions and would like to keep the rating.

---

### Official Review · Reviewer_RfAb · 2025-07-03

**Clarity:** 3
**Significance:** 3
**Originality:** 4
**Rating:** 5
**Confidence:** 3

**Summary:**

This paper studies the problem of detecting a dense bipartite subgraph. More specifically, this is a hypothesis testing problem between a null distribution of bipartite Erd\H{o}s–R\´enyi graphs (of size $n_1 \times n_2$) against an alternative distribution of bipartite random graph with a planted denser subgraph (of size $k_1 \times k_2$). The result establishes the minimax rate of the edge density separation $\delta^*$, for any values of $n_1, n_2, k_1$, and $k_2$. The optimal test consists of three parts: a total degree test, a truncated degree test, and a max truncated degree test. Each of them is optimal up to multiplicative constants under some conditions of the size parameters. A careful combination makes it optimal down to the complete regime, under a graph density constraint. It worth notice that the lower bound is not restricted on the graph density constraint.

**Questions:**

1. Could you clarify which specific regime in Assumption 1 is associated with Proposition 2?
2. Could you elaborate on the intuition for when the max-truncated degree test is preferable to the truncated degree test, and conversely, in which regimes the truncated degree test would be more appropriate? The construction of $\Delta_{a}^{h_1}$ and $\Delta_{a}^{h_2}$ have a simple threshold, but the other thresholds in terms of $\psi_{12}, \psi_{21}, \phi_{12},$ and $\phi_{21}$ are a bit hard to grab the idea.
3. Would you expect the lower bound to be tight when Assumption 1 fails?
4. A minor point I noticed that is the notation of $\Theta(n_1,n_2,k_1,k_2,\delta)$ in equation (1) is inconsistent with other places.

**Ethical Concerns:**

["NO or VERY MINOR ethics concerns only"]

**Final Justification:**

I raised my point as the authors addressed my main concerns in the rebuttal.

**Limitations:**

Yes.

**Paper Formatting Concerns:**

No concerns.

**Quality:**

3

**Strengths And Weaknesses:**

**Strengths**
1. This work gives a lower bound of the minimax rate of density separation $\delta^*$ under any $n_1,n_2,k_1,k_2$.
2. Although the upper bound result relies on a density constraint, it is very interesting to be independent of balanced community or balanced planted bipartite graph assumptions.
3. The proposed truncated degree test and max truncated degree test carefully extend the idea of the truncated $\chi^2$-test by coming up a statistic of the normalized (under null) average degree to control the tail probability.

**Weaknesses**
1. Many parts of the paper are well-written, but I find Section 3 a bit lack of intuitions. It would be nice to explain more on what would each regime look like when each of $\psi_{12}+\psi_{21}$, $\phi_{12}$, or $\phi_{21}$ becomes the smallest among those three quantities. This would largely help the readers grab the idea of when would each test become more natural or useful.
2. The constructed test potentially runs in exponential time, depending on the size of the planted dense bipartite subgraph. The potential of an information-computational gap does exist so we are not sure about whether this is a weakness of this constructed test or not. It would also be interesting to see whether there are some regimes where you can actually construct efficient tests.

---

> ### Author Rebuttal · Authors · 2025-07-30
>
> 1. *Could you clarify which specific regime in Assumption 1 is associated with Proposition 2?*
>
> Response: Proposition 2 is specifically associated with the third regime in Assumption 1, corresponding to $R = \phi_{21}$ and $n_1 > k_1^2$. We remark here that the form of Proposition 2 is in fact stronger than what is needed in Assumption 1; we include the result of Proposition 2 in the paper for the reader’s intuition. We will add a comment after Proposition 2 clarifying this in the camera-ready version of the paper.
>
> 2. *Could you elaborate on the intuition for when the max-truncated degree test is preferable to the truncated degree test, and conversely, in which regimes the truncated degree test would be more appropriate?*
>
> Response: In a nutshell, the truncated degree test is optimal when one of the two communities is large and the other is small. The max-truncated degree test is preferable when both communities are small.
>
> The intuition for the **truncated degree test** is as follows. Suppose that one of the communities is very large–for example, $k_2 = n_2$, to fix ideas– and the other one is small ($k_1 \ll n_1$). In the corresponding adjacency matrix, all entries within a row are iid Bernoulli random variables with the same probability parameter. It is well-known that a sufficient statistic of $m$ iid data points $X_1, \dots, X_m \sim Ber(q)$ is the sum $\sum_{i=1}^m X_i \sim Binomial(m,q)$. Therefore, the column vector obtained as the row-wise sum of the adjacency matrix is a sufficient statistic of the data. This vector is nothing but the vector of node degrees in the first population of the bipartite graph. Therefore, when one of the two communities is large enough, the problem can be reduced to a vector-based problem. A classical test statistic in such cases is the truncated chi-square test (see the definition of T at the bottom of Page 7), which we apply to the degree vector with a slight refinement to account for the subpoissonian tails of the binomial distribution. This yields the truncated degree test.
>
> The intuition for the **max-truncated-degree test** is as follows. When both communities are small, collapsing the matrix into a single vector (by summing over rows or columns) discards too much information from the data. In this case, a natural idea would rather be to scan over all possible bipartite subgraphs of size $k_1 \times k_2$ and reject the null hypothesis when one such subgraph contains an unusually large number of edges–an approach proposed in  (Rotenberg, Huleihel and Shayevitz, Planted Bipartite Graph Detection, IEEE TIT 2024). Unfortunately, this procedure is not optimal. Instead, we propose a non-trivial refinement that builds on the truncated chi-square test, adapted to tackle two key challenges: 1) the data are matrix-valued, not a vector-valued, and 2) the entries are Bernoulli rather than Gaussian random variables. Our procedure, formally defined in equation (14), is obtained by scanning and summing over subsets along one dimension of the matrix, applying a non-linear transformation based on the Bennett function, and truncating along the other dimension–analogously to the truncated chi-square test statistic. This procedure captures subtle concentration effects of bipartite graphs when both communities are small.
>
> Please let us know if this clarifies your intuition for when each test is optimal. If so, we will be happy to add this heuristic explanation to Section 3 of our paper to clarify the reader’s understanding.
>
>
> 3. *Would you expect the lower bound to be tight when Assumption 1 fails?*
>
> Response: We do not expect the lower bound to be tight when Assumption 1 fails. A similar gap has been established in the companion papers (Arias-Castro and Verzelen, Community detection in dense random networks, AoS 2014) and (Arias-Castro and Verzelen, Community detection in sparse random networks, AoS 2015), which study a special case of our setting by imposing the restriction $n_1 = n_2$ and $k_1 = k_2$. When the graph is sparse (that is, when an assumption on the connectivity akin to our Assumption 1 is violated), their lower bound requires further truncating the second moment of the likelihood ratio on a carefully chosen high-probability event, which leads to fundamentally different statistical rates.
>
> Here, we take a similar approach by focusing on dense graphs first and leaving the case of sparse graphs for future work. Our lower bound technique relies on a careful analysis of the moment generating function of the product of two binomial random variables. Assumption 1 is used in the upper bound only. Given the technical difficulty of the analysis under Assumption 1, we anticipate that relaxing this assumption will present substantial interesting challenges that we plan on pursuing in future work. We kindly refer the Referee to our response to Question 1 from Reviewer QZpt regarding possible extensions of our upper bounds when Assumption 1 does not hold.
>
> 4. *A minor point I noticed is that the notation of Theta in equation (1) is inconsistent with other places.*
>
> Response: We thank the reviewer for pointing out this typo, and we will fix this in the camera-ready version of our paper.
>
>
> *Regarding computational challenges:* We kindly refer the Referee to our response to Question 3 from Reviewer QZpt for a discussion of the computational challenges associated with our tests.

---

> > ### Comment · Reviewer_RfAb · 2025-08-06
> >
> > I thank the authors for the detailed response. The explanation on truncated degree test and max-truncated-degree test seems clear to me and it would be very helpful to include such discussion in the revised paper. I totally understand that the extension on violating Assumption 1 and the further investigation into the statistical-computational gap can be leaved for the future works. Based on these, I decide to raise my score to accept.
> >
> > Please make those edits accordingly in the camera-ready version.

---

### Note · Authors · 2025-08-13

Dear reviewers and Area Chair,

We would like to thank you all for a fruitful and stimulating discussion period. We believe that we have addressed all of the reviewers' comments to satisfaction, which has included clarifying the contribution of our work over existing literature, providing intuition for when each of our proposed tests is optimal, and perfecting our notation. This edits will greatly improve the quality of our work, and we hold that the final version of our paper will make a valuable contribution to the NeurIPS community.

Regards,
The authors

---

### Decision · Program_Chairs · 2025-09-17

**Decision:**

Accept (poster)

**Comment:**

The paper studies the problem of community detection in dense bipartite random graphs. More specifically, the authors analyze the following hypothesis testing problem: under the null hypothesis, the observed graph is drawn from a bipartite Erdős–Rényi distribution with connection probability $p_0$. Under the alternative hypothesis, there exists an unknown bipartite subgraph of size exactly $k_1 \times k_2$, in which edges appear with probability $p_1 = p_0 + \delta$, while all other edges outside the subgraph appear with probability $p_0$. The authors present minimax lower bounds and propose a test whose performance can be analyzed.

All reviewers acknowledged the soundness of the analysis. However, discussions in the rebuttal phase focused on assumptions, intuition, novelty, and computational challenges. These points should be incorporated into the paper. In particular:

The problem should be better motivated, as the authors currently do not mention any practical application.

The relevance of the problem should be discussed. At present, it seems rather narrow—partly because of the assumptions (dense regime), but mainly because of the objective itself: detecting whether there is a more connected set of nodes of size exactly $k_1 \times k_2$. Why is this specific formulation important?

The authors should provide a clearer explanation of why novel techniques are required for this problem, and explicitly outline the techniques they introduce.

The analysis is purely information-theoretic, meaning that computational aspects are not addressed. The authors should comment on the implementability of the proposed test and, if possible, include numerical experiments to support the theoretical findings.